# The impact of national suicide prevention strategies on suicide rates in the Region of the Americas: An interrupted time-series analysis using WHO Global Health Estimates

Shannon Lange[1], Katherine Guo[1] ![ORCID], Jurgen Rehm[1], Alessandra Trianni[2], Catalina Ortuzar-Lyon[2], Matias Irarrazaval[2], Carmen Martinez[2] and Renato Oliveira e Souza[2] ![ORCID]

[1]Centre for Addiction and Mental Health, Canada and [2]Pan American Health Organization, USA

## Research Article
National Suicide Prevention Strategies; Region of the Americas; Suicide; Time-series analysis

**Corresponding author:**
Shannon Lange;
Email: shannon.lange@camh.ca

A.T., C.O.-L., M.I., C.M. and R.O.S. are affiliated with the Pan American Health Organization. The authors declare sole responsibility for the views expressed in this article, which may not necessarily reflect the opinion or policy of the Pan American Health Organization/World Health Organization.

## Abstract

The development and implementation of national suicide prevention strategies (NSPSs) is one policy response to suicide prevention adopted by the World Health Organization (WHO); however, evidence on their effectiveness remains limited and mixed. This study assessed the impact of implementing an NSPS on sex-specific suicide mortality rates in nine countries within the Region of the Americas from 2000 to 2021. Suicide rates were obtained from the WHO Global Health Estimates, and countries with an NSPS and the year it was first implemented were identified using the WHO MiNDbank. A comparative interrupted time-series analysis using linear mixed-effects models was conducted to estimate the effect of NSPS implementation on suicide mortality. The implementation of an NSPS was associated with a gradual and sustained decrease in suicide mortality rates: 3.00% per year among males (95% CI: −5.28%, −0.66%) and 2.55% per year among females (95% CI: −4.62%, −0.44%). No significant difference in effect was observed between sexes. These findings demonstrate an association between NSPS and reduced suicide mortality in countries within the region, emphasizing the value of their ongoing development and implementation. Although the association did not vary by sex, NSPS design should account for sex-specific epidemiological contexts.

## Impact statement

This study shows that national suicide prevention strategies are associated with reductions in suicide mortality rates in countries within the Region of the Americas. Amid a global context in which the region stands out for increasing suicide rates, this research reinforces the importance of comprehensive national strategies as a critical public health intervention. The findings are relevant to policymakers, mental health practitioners and global health agencies aiming to reduce the burden of suicide through evidence-based strategies. Supporting the adoption and implementation of such strategies could accelerate progress toward global mental health and Sustainable Development Goals.

## Introduction

In recent years, attention to the prevention of suicide has mounted globally (UN General Assembly, 2015; World Health Organization, 2021a). For instance, the United Nations (UN) Sustainable Development Goals (SDGs), a set of international goals established in 2015 to improve global prosperity, included suicide as an indicator for Target 3.4: "By 2030, reduce by one third premature mortality from noncommunicable diseases through prevention and treatment and promote mental health and well-being" (UN General Assembly, 2015). Soon after, the World Health Organization (WHO) set an identical target in its Comprehensive Mental Health Action Plan (World Health Organization, 2021a). Despite these international goals to reduce the suicide mortality rate globally, when the WHO released its Global Health Estimates for 2019 (World Health Organization, 2021b), the suicide mortality rate in the Region of the Americas had an upward trend while decreasing in all other regions.

Accordingly, suicide is considered a serious public health problem in the Americas, which requires a strategic public health response. One way to address suicide is for countries to develop and implement a national suicide prevention strategy (NSPS). According to the World Health Organization (WHO), "a national strategy indicates a government's clear commitment to dealing with the issue of suicide" (World Health Organization, 2014). The

WHO outlines the key components of an NSPS as: clear object-ives, relevant risk and protective factors, effective interventions, prevention strategies at the general population level, prevention strategies for vulnerable sub-populations at risk, prevention strategies at the individual level, improving case registration and conducting research and monitoring and evaluation (World Health Organization, 2012).

The evidence base for the effectiveness of NSPS in reducing suicide is limited, and somewhat mixed (Taylor et al., 1997; De Leo and Evans, 2004; Matsubayashi and Ueda, 2011; Lewitzka et al., 2019; Schlichthorst et al., 2023). A recent study, which examined NSPS implementation and outcomes in 11 countries, further highlights the variation in strategy implementation and how evaluation continues to remain a key challenge within many of these strategies (Canal-Rivero et al., 2025). However, two of the largest and most recent multicountry analyses conducted suggest that, as a whole, NSPS are effective (Matsubayashi and Ueda, 2011), as opposed to any specific components (Schlichthorst et al., 2023). The study by Matsubayashi and Ueda (2011) com-prised data for 21 countries (11 with and 10 without an NSPS) belonging to the Organization for Economic Co-operation and Development for the years 1980–2004, and their regression model included country-specific political, economic and sociodemo-graphic covariates. The introduction of an NSPS was found to be associated with a significant decrease in the overall suicide rate, with a larger effect on the suicide rate among males compared to the rate among females. Schlichthorst et al. (2023) analyzed suicide data for 24 countries with an NSPS using interrupted time-series analysis. After adjusting for time trends, estimated period effects for overall suicide rates ranged from a significant decrease in the yearly suicide rate (RR = 0.80; 95% CI: 0.69–0.93) to a significant increase (RR = 1.12; 95% Confidence Interval [CI]: 1.05–1.19). There were no changes in suicide mortality associated with individual components of national strategies. Although these studies utilized data from more than 20 countries across the globe, there may be geographical variation in the impact of national suicide prevention strategies that were not accounted for. Further, the need for additional studies using robust evaluation designs, such as interrupted time-series analysis, has been acknowledged (Sinyor et al., 2024).

The objective of the current study was to evaluate the impact of implementing an NSPS on the sex-specific rate of suicide in the Region of the Americas. Given that such strategies are intended to initiate action, the implementation of an NSPS was hypothesized to be associated with a gradual, sustained decrease in the rate of suicide for both males and females. Although the literature suggests the impact may be heightened for males, compared with females, we refrained from deducing a sex-specific hypothesis as the litera-ture is limited.

## Methods

### Data

The WHO MiNDbank (https://extranet.who.int/mindbank/col lection/country) was used to ascertain which countries in the Region of the Americas had a comprehensive stand-alone NSPS and the year it was first implemented. To further corroborate the presence of an NSPS, the 2024 WHO Mental Health Atlas (World Health Organization, 2025), a triennial survey and report of countries' mental health policies and programs, laws, information systems, financing, workforce and services was used. The Atlas

data was obtained from a representative of the Ministries of Health or other responsible ministries in each given country. Given that a pre- and post-period was necessary, only those countries that developed an NSPS between 2000 and 2021 were included. That is, if a country had an NSPS throughout the study period or did not have an NSPS at any point within the study period, they were excluded. The United States of America was also excluded due to its limited pre-implementation period (an NSPS was first implemented in 2001). In the end, nine countries in the region were eligible for analysis: Argentina, Brazil, Chile, Domin-ican Republic, Guyana, Panama, Paraguay, Suriname and Uru-guay (Supplementary Table S1).

The most recent sex-specific age-standardized suicide mortality rate estimates (per 100,000 population) from the WHO Global Health Estimates 2000–2021 were retrieved, using the Pan Ameri-can Health Organization data portal (ENLACE) on noncommunic-able diseases, mental health, injuries and risk factors for each of the nine countries. The Global Health Estimates rely on statistical modeling to provide comprehensive and comparable estimates across countries by accounting for underreporting and misclassifi-cation. Rates are estimated by taking the total deaths by suicide by 5-year age groups, and sex are estimated for each country by applying the WHO life table death rates (World Health Organiza-tion, 2024b) to the estimated resident populations prepared by the United Nations Population Division in its 2024 revision (United Nations, 2024). In order to facilitate international comparability, age-standardized suicide mortality rates are computed using the WHO standard population (Ahmad et al., 2001), which assumes one standard age distribution of the population in all countries. For additional methodological details on the Global Health Estimates 2000–2021, see the respective Technical Paper (World Health Organization, 2024a).

### Statistical analysis

A comparative interrupted time-series analysis was conducted using linear mixed-effects models to examine the impact of implementing an NSPS on the sex-specific age-standardized sui-cide mortality rate, adjusting for year. The linear mixed-effects models used 11th-order autoregressive (AR(11); order chosen using autocorrelation plots) errors to account for serial autocor-relation within the data, as well as random intercepts for country. A dummy variable was created to capture the years an NSPS was in place, with 1 for the year it was established and for every year thereafter, and 0 for the years there was no strategy. Given that we hypothesized that implementing an NSPS would be associated with a gradual, sustained decrease in the suicide mortality rate for both males and females, a term representing the post-intervention slope (referred to as "time since NSPS" below) was included in each model. To determine if the impact was significantly different between the sexes, an interaction term between sex and post-intervention slope was included in a model using data for both males and females. Suicide mortality rates were log-transformed to reduce the level of skewness and stabilize the variance. Model results were converted to the percent-change scale to enhance interpretability, using the following formulas:

$$\text{Point estimate} : 100 * (\exp(\text{ß}) - 1)$$

$$\text{Standard error} : 100 * \text{SE} * \exp(\text{ß})$$

$$95\% \text{CI} : 100 * [\exp(\text{ß} \pm 1.96 * \text{SE}) - 1]$$

Model assumptions related to linearity and homoskedasticity were verified using residual QQ plots (Supplementary Figures S1, S4 and S7) and residual versus fitted plots (Supplementary Figures S2, S5 and S8). Although it is somewhat challenging to assess for violations of the homoskedasticity assumption, given the relatively low number of data points available, mixed-effect models with continuous outcomes are fairly robust to violations of distributional assumptions (Schielzeth et al., 2020). Autocorrelation plots were also used to verify that no underlying serial autocorrelation remained within the models (Supplementary Figures S3, S6 and S9). Variance inflation factor (VIF) was used to detect multicollinearity in the analyses, with a threshold of 10 used to indicate its presence. Statistical significance was set at $\alpha \leq 0.05$. All statistical analyses were conducted using R version 4.4.3.

### Sensitivity analyses

As specified above, an 11th-order model was chosen based on the autocorrelation plots. A sensitivity analysis was conducted using a simpler error structure (i.e., a first-order model) to determine whether the use of such a complex error structure impacted model estimation, given the relatively limited time-series available. Further, given concerns raised in the current literature about the implementation of an NSPS in Argentina and Brazil (Barcala and Faraone, 2023; Canal-Rivero et al., 2025; Wille Augustin et al., 2025), a second sensitivity analysis removing data from Argentina and Brazil from the analytical sample was conducted.

### Results

In total, there were 198 observations for each sex across the nine countries included in the analyses. The rate of suicide among males was higher compared with females in all nine countries. The suicide mortality rate ranged from 5.6 (95% uncertainty interval [UI]: 4.2, 7.2; in Panama) to 43.9 (95% UI: 31.5, 59.9; in Guyana) among males, and from 1.0 (95% UI: 0.8, 1.2; in Panama) to 14.2 (95% UI: 9.0, 21.0; in Suriname) among females, in 2021 (the latest available year). The mean suicide mortality rate across study years for the nine countries combined was 5.4 per 100,000 (SD = 4.4) among females and 20.3 per 100,000 (SD = 15.3) among males. Figure 1 displays the sex-specific age-standardized suicide mortality rate for each of the nine included countries from 2000 to 2021, as well as the years in which an NSPS was implemented within each country.

The model results suggest that NSPS had a gradual, sustained impact on the suicide mortality rate for both males and females. Specifically, for every year after the implementation of an NSPS there was, on average, a 3.00% (95% CI: −5.28%, −0.66%, $p$-value = 0.013) decrease in the suicide mortality rate among males and a 2.55% (95% CI: −4.62%, −0.44%, $p$-value = 0.019) decrease in the suicide mortality rate among females, compared to previous years. The interaction term between sex and post-intervention slope in the model using the data for both males and females indicates that the impact was not significantly different between the sexes ($p$-value = 0.346; Table 2). The VIF indicated that there was minimal multicollinearity present in the models. Conditional

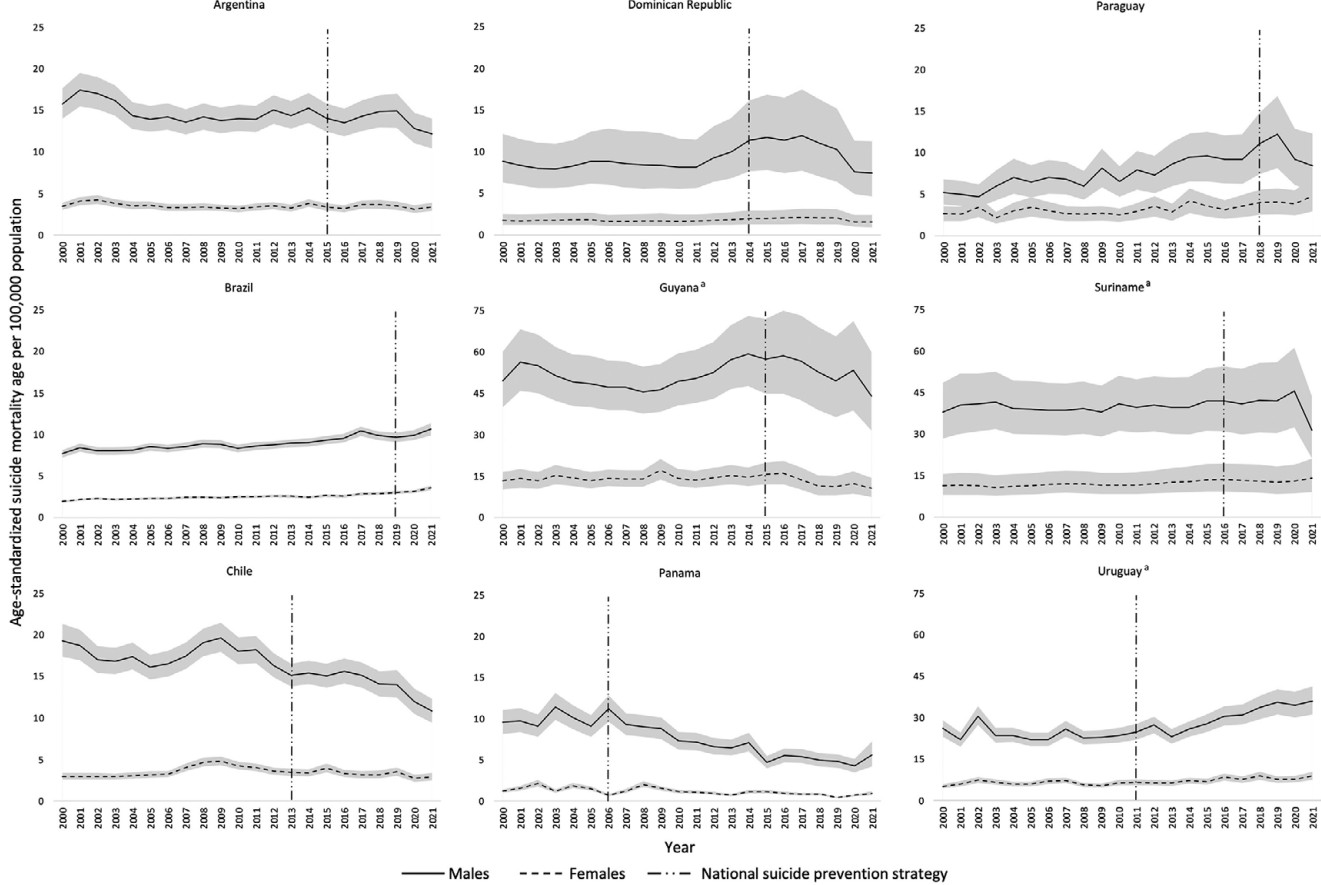

**Figure 1.** Age-standardized suicide mortality rate by sex from 2000 to 2001 and the year in which a National Suicide Prevention Strategy was implemented for each of the nine countries. *Note*: [a] It should be noted that the scale for Guyana, Suriname and Uruguay goes notably higher than those for the other countries included in the analyses, as the suicide mortality rate in these three countries is among the highest in the region.

$R^2$ values were above 0.90 across all models, meaning that over 90% of the variance in the data was explained by the models.

### Sensitivity analyses

Both sensitivity analyses resulted in interpretations similar to those of our main analysis. The use of AR(1) errors produced point estimates, which were slightly more extreme than those produced using AR(11) errors for both sexes (males: 3.37% decrease, 95% CI: −5.74%, −0.94%, *p*-value = 0.008; females: 3.67% decrease, 95% CI: −5.55, −1.77, *p*-value <0.001; see Table 1). There was also no evidence of a sex-interaction when using AR(1) errors (*p* = 0.487; Table 2). The exclusion of Argentina and Brazil resulted in point estimates, which were slightly higher than those produced using the analytical dataset sample, including these respective countries (males: 3.16% decrease, 95% CI: −6.00, −0.24, *p*-value = 0.036; females: 3.01% decrease, 95% CI: −5.45, −0.51, *p*-value = 0.020). There was, once again, no evidence of a sex interaction (*p* = 0.444).

### Discussion

The current analyses found that the development and implementation of an NSPS is associated with a gradual, sustained decrease in the suicide mortality rate for both males and females, supporting the overall hypothesis. Although a sex difference has been previously reported, with a significant impact found for males but not for females (Matsubayashi and Ueda, 2011), this was not the case in the current study. The lack of sex difference found may be a reflection of well-rounded strategies that include effective, planned, targeted actions for each sex, which should be a goal of any NSPS. The lack of difference may also be reflective of differences in strategy

implementation and/or the specific populations included within this study, as they may somehow differ systematically from previously studied populations. The study by Matsubayashi and Ueda (2011) indicates that some of their included countries possessed strategies specifically targeted toward one particular sex. The investigators also relied on the observed differences in magnitude and statistical significance to draw their conclusions, rather than formally testing for a difference. However, such methods of assessment are not always ideal, and the difference between significant and nonsignificant is itself not necessarily statistically significantly different (Gelman and Stern, 2006). Having said that, it has been found that the contextual factors that significantly impact the suicide mortality rate among males and females in the Region of the Americas are largely different, supporting the fact that sex should be considered when adapting and testing suicide risk reduction interventions, and when developing an NSPS in a country (Lange et al., 2023). Therefore, despite not finding a sex difference in the impact of an NSPS on suicide mortality, it is still necessary to consider sex differences in the epidemiology of suicide, as well as in prevention interventions during the development of an NSPS.

The components of the NSPS that were inherently evaluated here were not compared or contrasted, as it is expected that strategies will vary depending on the epidemiology of suicide and sociocultural contexts within each country. However, a recent study that strived to identify which specific components of an NSPS were associated with a decrease in suicide mortality did not find evidence of any specific contributions (Schlichthorst et al., 2023). It is possible that it is a strategy in its entirety that is effective (i.e., associated with a statistically significant decrease in the suicide mortality rate) rather than one particular component. This notion is further supported by the evidence for multicomponent interventions and their potential

**Table 1.** Sex-specific comparative interrupted time-series analysis results

| | Males | | | | | | | Females | | | | | | |
| | | 95% CI | | | | 95% CI | | | 95% CI | | | | 95% CI | |
| Variable | effect | Lower | Upper | *p*-value | VIF | Lower | Upper | effect | Lower | Upper | *p*-value | VIF | Lower | Upper |
|---|---|---|---|---|---|---|---|---|---|---|---|---|---|---|
| **Main analysis, AR(11) model** | | | | | | | | | | | | | | |
| Intercept | −100.00 | −100.00 | 3.19x10⁹ | 0.487 | – | – | – | −100.00 | −100.00 | 595.71 | 0.072 | – | – | – |
| Year | 0.62 | −0.72 | 1.97 | 0.369 | 1.88 | 1.58 | 2.35 | 1.25 | −0.03 | 2.54 | 0.057 | 1.80 | 1.52 | 2.25 |
| Time since NSPS[a] | −3.00 | −5.28 | −0.66 | 0.013 | 1.88 | 1.58 | 2.35 | −2.55 | −4.62 | −0.44 | 0.019 | 1.80 | 1.52 | 2.25 |
| $R^2$ | 0.919 | – | – | – | – | – | – | 0.933 | – | – | – | – | – | – |
| **Sensitivity analysis, AR(1) model** | | | | | | | | | | | | | | |
| Intercept | −100.00 | −100.00 | 4.54x10⁵ | 0.191 | – | – | – | −100.00 | −100.00 | −100.00 | <0.001 | – | – | – |
| Year | 0.99 | −0.28 | 2.29 | 0.128 | 2.06 | 1.71 | 2.58 | 1.68 | 0.76 | 2.60 | <0.001 | 2.31 | 1.90 | 2.90 |
| Time since NSPS[a] | −3.37 | −5.74 | −0.94 | 0.008 | 2.06 | 1.71 | 2.58 | −3.67 | −5.55 | −1.77 | <0.001 | 2.31 | 1.90 | 2.90 |
| $R^2$ | 0.921 | – | – | – | – | – | – | 0.957 | – | – | – | – | – | – |
| **Sensitivity analysis, excluding Argentina and Brazil** | | | | | | | | | | | | | | |
| Intercept | −100.00 | −100.00 | 1.50x10¹³ | 0.620 | – | – | – | −100.00 | −100.00 | 5,467.29 | 0.092 | – | – | – |
| Year | 0.58 | −1.13 | 2.32 | 0.510 | 2.19 | 1.77 | 2.83 | 1.37 | −0.13 | 2.89 | 0.075 | 2.28 | 1.84 | 2.96 |
| Time since NSPS[a] | −3.16 | −6.00 | −0.24 | 0.036 | 2.19 | 1.77 | 2.83 | −3.01 | −5.45 | −0.51 | 0.020 | 2.28 | 1.84 | 2.96 |
| $R^2$ | 0.933 | – | – | – | – | – | – | 0.954 | – | – | – | – | – | – |

[a]Post-intervention slope.
NSPS, National suicide prevention strategy.

**Table 2.** Comparative interrupted time-series analysis results with interaction between post-intervention slope and sex

| Variable | Effect | 95% CI | | p-value | VIF | 95% CI | |
|---|---|---|---|---|---|---|---|
| | | Lower | Upper | | | Lower | Upper |
| **Main analysis, AR(11) model** | | | | | | | |
| Intercept | −100.00 | −100.00 | −99.81 | 0.008 | – | | |
| Year | 1.26 | 0.38 | 2.15 | 0.005 | 2.15 | 1.87 | 2.51 |
| Time since NSPS[a] | −2.79 | −4.67 | −0.88 | 0.005 | 3.11 | 2.67 | 3.69 |
| Sex, male | 312.11 | 229.28 | 415.77 | <0.001 | 1.09 | 1.02 | 1.32 |
| Time since NSPS*Sex | −1.04 | −3.17 | 1.14 | 0.346 | 2.05 | 1.79 | 2.40 |
| $R^2$ | 0.978 | | | | | | |
| **Sensitivity analysis, AR(1) model** | | | | | | | |
| Intercept | −100.00 | −100.00 | −100.00 | <0.001 | – | | |
| Year | 1.49 | 0.76 | 2.23 | <0.001 | 2.30 | 1.99 | 2.70 |
| Time since NSPS[a] | −3.29 | −5.05 | −1.50 | <0.001 | 3.27 | 2.79 | 3.87 |
| Sex, male | 305.86 | 229.68 | 399.65 | <0.001 | 1.05 | 1.01 | 1.43 |
| Time since NSPS*Sex | −0.71 | −2.67 | 1.30 | 0.487 | 2.02 | 1.77 | 2.36 |
| $R^2$ | 0.980 | | | | | | |
| **Sensitivity analysis, excluding Argentina and Brazil** | | | | | | | |
| Intercept | −100.00 | −100.00 | −97.64 | 0.023 | – | – | – |
| Year | 1.37 | 0.26 | 2.48 | 0.016 | 2.51 | 2.13 | 3.01 |
| Time since NSPS[a] | −3.11 | −5.32 | −0.85 | 0.008 | 3.48 | 2.91 | 4.23 |
| Sex, male | 317.96 | 214.49 | 455.48 | <0.001 | 1.08 | 1.02 | 1.39 |
| Time since NSPS*Sex | −0.95 | −3.33 | 1.50 | 0.444 | 2.05 | 1.76 | 2.45 |
| $R^2$ | 0.978 | – | – | – | – | – | – |

[a]Post-intervention slope.
NSPS, National suicide prevention strategy.

synergistic effect on suicide rates (Ono et al., 2013; Mergl et al., 2023; Shand et al., 2025). Regardless, the likely differences in the components included in each NSPS highlight the need for future country-specific studies, which may only be possible for some after sufficient time has passed. Beyond the statistical reasoning (Jiang et al., 2022), the importance of sufficient time is further strengthened by the possibility that improvements in the monitoring and registration of cases of death by suicide that could come with the implementation of an NSPS in some countries, may also result in an apparent increase in the suicide mortality rate.

It should be acknowledged that the efficacy of such strategies is dependent on the degree to which they are implemented. Although evaluating the stage of implementation was beyond the scope of the current paper, as reported elsewhere (Canal-Rivero et al., 2025), it is likely that implementation varied across the countries included here. The findings of the sensitivity analysis excluding Argentina and Brazil due to concerns surrounding the execution of their NSPS lend support to this belief, as the point estimates increased slightly for both males and females, compared to the main analysis; possibly a reflection of the stage of implementation of the NSPS in these countries.

In contrast to previous studies that tested the immediate impact of an NSPS on suicide mortality, the current analysis tested a gradual, sustained impact. This was done in an effort to account for the complex process of implementing an NSPS over time (Platt et al., 2019). Additional strengths of the current study include the use of the most up-to-date suicide mortality rate estimates, which span from 2000 to 2021, and the multicountry approach. The multi-country approach improved internal validity, enhanced external validity, increased statistical power and precision, and avoided issues with unbalanced time-series (i.e., wherein the intervention did not occur near the mid-point of the time series) (Jiang et al., 2022). However, it should be acknowledged that the time series available precluded the inclusion of countries that had an NSPS prior to 2000/post-2021 or insufficient pre- or post-implementation time points, and the multicountry approach eliminated the ability to obtain country-specific estimates. It may be the case that country-specific findings differ from the overall finding reported here. Additional limitations include the inability to include potentially relevant contextual factors, which may have had an impact on suicide mortality rates, due to limited degrees of freedom and the increased likelihood of multicollinearity. For instance, the UN SDGs released in 2015 marked a potential shift in attention to suicide prevention. However, multiple countries included in the current analyses had enacted an NSPS at a similar point in time, with two of the countries having implemented an NSPS in 2015 exactly; therefore, due to concerns with multicollinearity, it was not possible to include the release of the SDGs as a covariate. In a similar vein, given the nature of the analysis and lack of adjustment for confounding, causality cannot be inferred. However, the fact that the association was still observed after averaging across multiple countries, despite differences in implementation and other contextual factors, may lend greater credence to the

notion of a true impact of NSPS. Further, regression to the mean cannot be ruled out, as an NSPS is more likely to be developed and implemented in response to high, increasing suicide mortality rates. Although an NSPS had been implemented in three countries among those with the highest suicide mortality rates in the Region of the Americas (Pan American Health Organization (PAHO), 2021), there were also a number of countries included that had much lower suicide mortality rates and yet still implemented an NSPS (as shown in Figure 1). Lastly, it must be acknowledged that the current study constitutes an outcome evaluation in the narrowest possible sense, meaning that the prevention of death by suicide is only one of the outcomes that can be expected with the implementation of an NSPS. Other ancillary outcomes, such as other indicators of mental distress as well as suicide attempts, should be considered in effectiveness evaluations.

## Conclusion

The findings of the current study provide evidence of an association, among countries within the Region of the Americas, between NSPS and gradual declines in suicide mortality rates over time, with a similar pattern observed across the sexes. At a time when suicide mortality rates have been increasing in the region, the adoption and support of suicide prevention interventions remain important. Countries within the region, and elsewhere, should continue to be encouraged to develop and implement a comprehensive NSPS.

**Open peer review.** To view the open peer review materials for this article, please visit http://doi.org/10.1017/gmh.2026.10173.

**Supplementary material.** The supplementary material for this article can be found at http://doi.org/10.1017/gmh.2026.10173.

**Data availability statement.** The data used in the current study are publicly available and can be obtained from their original sources.

**Author contribution.** Conceptualization: S.L., R.O.E.S.; Data interpretation: S.L., J.R.; Formal analysis: K.G.; Methodology: S.L., J.R.; Supervision: R.O.E.S.; Validation: A.T., C.O.-L., M.I., C.M.; Visualization: S.L.; Writing – original draft: S.L., K.G.; Writing – review and editing: J.R., A.T., C.O.-L., M.I., C.M., R.O.E.S. All authors approved the final version of the manuscript.

**Competing interests.** The authors declare none.

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
