## [Reviewer Report]

This is an important study to help inform efforts across the globe to develop and implement national suicide prevention strategies.Such evidence is important to help persuade national policymakers to support national programs aimed at reducing suicide.This is particularly important in the Americas where there has been an upward trend while decreasing in all other regions.Of particular importance was utilizing a methodology that could detect a gradual sustained impact given the time needed for implementation of a National Strategy. Also of importance was the finding that it may be the strategy in its entirety that is effective rather than any individual components.

---

## [Reviewer Report]

Dear Editor,

I have reviewed the manuscript GMH-2025-0222, titled “The Impact of National Suicide Prevention Strategies on the Rate of Suicide in the Region of the Americas: An Interrupted Time-Series Analysis.” The study addresses an important and highly relevant topic for global public health, especially in the Region of the Americas, where suicide rates have been increasing. The results suggesting an association between the implementation of National Suicide Prevention Strategies (NSPS) and a gradual and sustained decline in suicide mortality rates are valuable.

However, in its current state, the manuscript presents several areas that require substantial improvement in theoretical foundation, methodological clarity, and the depth of the discussion and conclusions.

General and Major Comments

1. Update of the Literature Review: The Introduction and Discussion sections need to be updated with the most recent and relevant literature regarding the effect of national suicide prevention plans. It is crucial for the authors to incorporate and critically discuss all recent findings that examine the effectiveness of NSPS to adequately contextualize their results.

2. Verification of NSPS Authenticity (Argentina and Brazil): I have serious doubts about the correct classification of policies in Argentina and Brazil as true NSPS, given that recent publications have questioned whether some national policies meet the requirements to be considered an integral and autonomous suicide prevention strategy (NSPS). The authors must comprehensively review and justify in the Methods section and/or an Appendix whether the policies of these countries, according to the WHO MiNDbank database and the WHO’s definition, genuinely constitute a comprehensive and differentiated NSPS. This justification is critical, as the validity of the study’s main intervention relies on this classification.

3. NSPS Implementation and Reporting: The authors note that a gradual impact is expected due to the “complex process of implementing an NSPS over time.” However, the manuscript omits key information about whether the national plans have actually been implemented in the countries analyzed. It is vital to include:

• A statement or appendix detailing the implementation status of the NSPS for all nine countries.

• Reports (if available) on whether the national plans have been implemented. Reports exist for several countries mentioned by the authors, such as Chile, and this information should be incorporated to validate the start of the strategy’s true implementation phase.

4. Methodological Limitations and Statistical Assumptions: The Methods section requires significant expansion to ensure the robustness of the analysis:

• ITS Model Requirements: The authors must detail whether all necessary statistical assumptions (e.g., the normality assumption for residuals) for the application of an Interrupted Time-Series (ITS) analysis with linear mixed-effects models, including the use of AR(1) errors, were met.

• Grouping of Countries in ITS: The use of a single ITS model that groups all countries (with adjustment for random effects per country) raises methodological concerns. Given that the NSPS were implemented in different years and the epidemiological and policy contexts vary widely, the authors must provide a more robust justification for why aggregating all countries into a single comparative ITS model is the most appropriate methodological approach and how the risks of aggregation bias or heterogeneity are mitigated.

5. Definition of Key Concepts: In the text (possibly the Introduction or Discussion), the acronym UN SDGs (United Nations Sustainable Development Goals) must be clearly explained the first time it is mentioned.

6. Discussion and Conclusion:

• Discussion Improvement: The Discussion section is currently poor and too general. The authors need to:

o Include and discuss the specific information available regarding the components of the NSPS implemented in the nine countries, acknowledging that strategies may vary. If available, they should discuss how these variations might influence the results, even if previous studies suggest the effect might be due to the strategy as a whole.

o Elaborate on the potential reasons for the results they found, such as why they did not find a significant difference in impact between sexes, despite previous literature suggesting a potentially greater impact on males.

o Delve deeper into the practical implication of the non-difference in impact by sex result, while maintaining the need to address the specific epidemiological contexts of each sex.

• Conclusion Improvement: The conclusion is poor and general. Given that the methodology employed—an ITS with mixed effects and follow-up only until 2021—may have limitations in establishing a direct causal relationship, the conclusions must reflect the actual strength of the results. The final conclusion should be rewritten to be more nuanced and directly supported by the specific results of the ITS model. For instance, instead of a categorical statement, it should emphasize the association found and the importance of the adoption and continuation of these strategies in the region.

Thank you for the opportunity to review this paper.

---

## [Reviewer Report]

This is an excellent, clearly written paper with crucial policy implications for suicide prevention and important contributions to the evidence base. Results show, on average, reductions in suicide following implementation of national suicide prevention strategies (NSPS). The authors thus conclude that countries should continue to develop and implement national suicide prevention strategies in the Americas.

Please consider the following:

1. Introduction: The WHO target in the Comprehensive Mental Health Action Plan was updated to a 1/3 reduction by 2030 – see Pg. 4, Introduction for revision: “Soon after, the World Health Organization (WHO) set an explicit target of a 10% reduction in the rate of suicide in countries by 2030 in their Comprehensive Mental Health Action Plan (World Health Organization 2021a).”

2. Introduction/Discussion: Suggest acknowledging that suicide rates may increase following NSPS due to enhanced focus on surveillance and strengthening of suicide case registration.

3. Introduction/Discussion: Suggest including the evidence for multi-component interventions and their potential synergistic effect on suicide rates, particularly after the following sentence in the discussion: “Thus, it is possible that it is a strategy in its entirety that is effective (i.e., associated with a statistically significant decrease in the suicide mortality rate) rather than one particular component.” See:

a. Shand F, Torok M, Mackinnon A, Burnett A, Sharwood LN, Batterham PJ, Calear AL, Qian J, Zeritis S, Sara G, Page A. Effect of the LifeSpan suicide prevention model on self-harm and suicide in four communities in New South Wales, Australia: a stepped-wedge, cluster randomised controlled trial. BMJ mental health. 2025 Mar 31;28(1).

b. Ono Y, Sakai A, Otsuka K, Uda H, Oyama H, Ishizuka N, Awata S, Ishida Y, Iwasa H, Kamei Y, Motohashi Y. Effectiveness of a multimodal community intervention program to prevent suicide and suicide attempts: a quasi-experimental study. PLoS One. 2013 Oct 9;8(10):e74902.

c. Mergl R, Heinz I, Allgaier AK, Hegerl U. Munich Alliance Against Depression: Effects of a Community-Based Four-Level Intervention Program on Suicide Rates. Crisis. 2022 Jun 27.

4. Methods/Results: Given the limited number of countries in the study, it would be interesting to analyse the degree of variation in impact between countries, and in which countries the impact was greatest and where it was less. This would help to understand potential enablers and barriers.

5. Methods/Discussion: It should be acknowledged that the data from some of these countries rely on modelled estimates.

---

## [Editor Report]

Dear Dr Lange and colleagues,

Thank you for your submission. Following receiving two vastly different recommendations, I sent it out for another - which has worked out in your favour. 

I am keen on reading your next iteration. Please carefully address the reviews, responding to each point raised by our reviewers.

Thank you and all the best,

Dr Sandersan Onie

---

## [Reviewer Report]

I thank the authors for addressing the comments. The paper is a meaningful contribution to the field and will help support strategic action and leadership for suicide prevention in countries.

---

## [Editor Report]

Dear Prof Large and colleagues,

Thank you for taking the time to carefully revise this manuscript. It has been strengthened substantially through the revision process. I have a few minor comments that I hope you will consider implementing:

1. It would be helpful for readers to better understand the limitations of the paper through clearer identification of the data source. As such, please revise the title and abstract to specify the WHO Global Health Estimates as the source of mortality data.

2. Please also mention the use of MiNDbank in the abstract to clarify how countries with a national suicide prevention strategy were identified, including when each strategy commenced.

3. I share similar sentiments to Reviewer 2 in that it is very difficult to ascertain whether national suicide prevention strategies were responsible for the effects observed. This is not a weakness of the paper, but rather a natural consequence of studying this critically important phenomenon. However, I suggest that the conclusions be tempered in both the abstract and the main text, avoiding causal language—particularly in light of the limitations outlined in the manuscript and the fact that detailed implementation analyses are beyond the scope of this study.

Once again, thank you for this important and timely work.

Kind regards,

Sandy

---

## [Editor Report]

Dear Prof Lange and colleagues,

Thank you for taking the time to revise the manuscript. I am happy to recommend it for publication.

Congratulations on this important contribution to the literature. 

All the best,

Dr. Sandersan Onie